# Bacterial Community Survey of *Wolbachia*-Infected Parthenogenetic Parasitoid *Trichogramma pretiosum* (Hymenoptera: Trichogrammatidae) Treated with Antibiotics and High Temperature

**DOI:** 10.3390/ijms24098448

**Published:** 2023-05-08

**Authors:** Wei Guo, Meijiao Zhang, Liangguan Lin, Chenxu Zeng, Yuping Zhang, Xiaofang He

**Affiliations:** 1Department of Entomology, College of Plant Protection, South China Agricultural University, Guangzhou 510642, China; 2Institute of Plant Protection, Guangdong Academy of Agricultural Sciences, Guangzhou 510640, China; 3Key Laboratory of Bio-Pesticide Innovation and Application, Guangzhou 510642, China

**Keywords:** *Trichogramma pretiosum*, *Wolbachia*, 16S rRNA, reproductive manipulate

## Abstract

*Wolbachia* has been shown to induce thelytokous parthenogenesis in *Trichogramma* species, which have been widely used as biological control agents around the world. Little is known about the changes of bacterial community after restoring arrhenotokous or bisexual reproduction in the *T. pretiosum*. Here, we investigate the emergence of males of *T. pretiosum* through curing experiments (antibiotics and high temperature), crossing experiments, and high-throughput 16S ribosomal RNA sequencing (rRNA-seq). The results of curing experiments showed that both antibiotics and high temperatures could cause the thelytokous *T. pretiosum* to produce male offspring. *Wolbachia* was dominant in the thelytokous *T. pretiosum* bacterial community with 99.01% relative abundance. With the relative abundance of *Wolbachia* being depleted by antibiotics, the diversity and relative content of other endosymbiotic bacteria increased, and the reproductive mode reverted from thelytoky to arrhenotoky in *T. pretiosum*. Although antibiotics did not eliminate *Wolbachia* in *T. pretiosum*, sulfadiazine showed an advantage in restoring entirely arrhenotokous and successive bisexual reproduction. This study was the first to demonstrate the bacterial communities in parthenogenetic *Trichogramma* before and after antibiotics or high-temperature treatment. Our findings supported the hypothesis that *Wolbachia* titer-dependence drives a reproduction switch in *T. pretiosum* between thelytoky and arrhenotoky.

## 1. Introduction

In the haplo-diploid sex determination system, females develop from fertilized diploid eggs through bisexual reproduction, whereas males are produced from unfertilized haploid eggs through arrhenotoky. Thelytokous parthenogenesis, in which females develop from unfertilized diploid eggs, is particularly common in solitary Hymenoptera. Causes of thelytokous probably occur on a nuclear genetic basis or are induced by cytoplasmic bacteria belonging to the Rickettsiaceae family and the *Wolbachia* genus [1].

*Trichogramma* (Hymenoptera: Trichogrammatidae) wasps are minute egg parasitoids with the most research and the widest application range for biological control, which have achieved the most remarkable economic benefits in the world [2]. In general, the main reproductive mode of wasps is bisexual reproduction (or arrhenotoky), while certain strains of wasps contain thelytokous females, which produce female offspring from unfertilized eggs. Compared with the bisexual line of *Trichogramma*, thelytokous parthenogenesis has certain advantages: (1) Their population is composed of females, which can reduce the cost of mass rearing and the quantity of wasps to release in fields when used as a biological control agent. (2) As a result of thelytokous, it promotes the establishment and spread of *Trichogramma* populations in the field [3].

It has previously been reported that endosymbiotic bacteria induce thelytokous parthenogenesis in *Trichogramma* species [4]. There are as many as 18 species of *Trichogramma* infected with parthenogenesis-inducing (PI) *Wolbachia*, accounting for at least 9% of the total *Trichogramma* [5]. To explore the functions of *Wolbachia* on insects, it is essential to efficiently obtain enough uninfected individuals or phenotypes changed from thelytoky to arrhenotoky. Typically, antibiotics and high temperature are used to deplete *Wolbachia* titer from their hosts and induce thelytoky to revert to arrhenotokous/bisexual reproduction [4,6,7,8,9,10,11,12,13,14,15,16]. Reducing the *Wolbachia* titer increases the proportion of males in the offspring. However, most antibiotics, including tetracycline hydrochloride, increased the proportion of male offspring and failed to eliminate *Wolbachia* infection [17,18,19]. In clinical trials, it was shown that it takes 1–2 years to deplete *Wolbachia* [20,21]. Moreover, the *Wolbachia* titer rebounded and embryogenesis returned to normal following antibiotic treatment [22]. However, how these processes are induced and regulated remains enigmatic.

Microbiota either provide the host insects with the necessary nutrients for growth or affect their adaptability and manipulate host reproduction, such as *Wolbachia* [4]. Therefore, studying the interaction between insects and microbiota is of great importance for studying the coevolution between insects and microbiota [23,24]. The 16S rRNA-seq of metagenome can be used to identify dominant microorganisms present in biological samples, and it is an accurate method to detect microbial infection without culture [25]. Most of the previous studies have investigated the interaction between insects and gut microflora through 16S rRNA-seq, such as cicadas [26], mosquitoes [27], and fruit flies [28]. The reproductive mode of *T. pretiosum* infected with PI *Wolbachia* will be changed to varying degrees by exposure to antibiotics and high temperatures [4,16]. We hypothesize that the bacterial community and the core microflora have different potential capacities to respond to the antibiotics and high temperature leading to the changing of reproductive modes in *T. pretiosum.* Additionally, comparative analysis of *Trichogramma* microbiomes from different treatment based on 16S rRNA-seq and the relationship to reproductive regulation has yet to be conducted.

In this study, we explored the causes of thelytoky in the *T. pretiosum*, using antibiotics (sulfadiazine, tetracycline hydrochloride) and high temperature. An effective approach to establishing entire arrhenotoky of *T. pretiosum* was screened out by curing experiments and a crossing experiment. The diversity and relative abundance changes of endosymbiotic bacterium with *Wolbachia* as the core microbial group were detected by 16S rRNA-seq. The findings of this study support the dependence on *Wolbachia* titer of manipulating the host’s thelytokous reproduction and the first to demonstrate the bacterial communities in parthenogenetic *Trichogramma* before and after antibiotics and high-temperature treatment. Compared with other experimental methods, 16S rRNA-seq can simultaneously assess the microbial diversity and functional activity potential of each taxon. The research on the bacterial community of *T. pretiosum* will help us understand the core microflora and their responses to environmental changes through the change of productive modes.

## 2. Results

### 2.1. Reversion to Bisexual Reproduction in T. pretiosum Exposed to Sulfadiazine

The bisexual reproduction line was completely restored after two generations of sulfadiazine treatment of single female segregation and crossing with male offspring. The biological characteristics of *T. pretiosum* after antibiotics and high-temperature treatments are shown in Table 1. Female (F1) produced 100% male offspring after rearing with sulfadiazine for two generations, whether successively fed sulfadiazine or honey water (*p* < 0.001). Tetracycline hydrochloride and high temperature did not entirely change the reproductive mode (partly some thelytokous parthenogenesis) of *T. pretiosum*.

When treated with sulfadiazine, the number of parasitic eggs showed no significant differences in the five generations compared with the control. The emergence rate was higher than the control group (F0, F1(S): *p* < 0.001; F1(H), F2(S), F2(H): *p* < 0.05), a probable cause being super parasitism in the sulfadiazine treatment. After cross-generational experiments (F1 female and F2 male), the offspring with male and female were produced, and the proportion of male was about 30% (*p* < 0.001). The frequency of *Wolbachia* infection in the treatment group of F1 and F2 (10%, 0%) with *T. pretiosum* females was significantly lower (*p* < 0.001) than for the control group (100%). No intersexual individuals were produced in the sulfadiazine treatment. Both sexes appeared when males (F2S or F2H) from “reverted” lines were completely compatible in crosses with females (F1). PCR assay failed to detect *Wolbachia* in the cross F3–F6 generations.

When treated with tetracycline hydrochloride, the number of parasitic eggs and emergence rate were significant lower found during the following first five generations compared with the control group (Parasitic eggs: F0, F1, F3: *p* < 0.001; F2, F4: *p* < 0.05; Emergence rate: F0, F3, F4: *p* < 0.05, F1, F2: *p* < 0.001). Though males appeared when rearing with tetracycline hydrochloride, the male percentage was lower than that of female (*p* < 0.001).

When treated with high temperature, the number of parasitic eggs reduced significantly in the first generation (F0: *p* < 0.001), and then increasing to higher than control in the following generations (F2, F3, F4: *p* < 0.001). The emergence rate tended to decrease in the first five generations when compared to the control. The emergence rate was significantly lower than that of control in the third and fourth generations (*p* < 0.05). The male percentage tended to increase by the first five generations. *Wolbachia* infection was 100% detected and intersex individuals appeared after tetracycline hydrochloride and high-temperature treatment.

### 2.2. Bacterial Diversity Analysis in T. pretiosum

To investigate the microbiome associated with *T. pretiosum* that were treated with honey water, two antibiotics, and high temperature, a total of 15 samples were subjected to 16S rRNA-seq analysis using the Ion Torrent PGM™ sequencing platform. The total obtained reads accounted for 321,747 (Appendix A). They were filtered for chimeras, quality score, and copy number. Reads were classified according to 16S rRNA variable regions for each sample and only reads with a large number of copies were mapped against 16S rRNA databases.

The distribution of shared and unique OTUs among bacterial communities in the studied samples is presented in Figure 1 (Appendix A). The number of exclusive taxa was the most diverse (220 taxa) for males produced by feeding sulfadiazine (TPW-S-M) followed by females produced by feeding sulfadiazine (TPW-S-F, 157 taxa). In comparison, females produced by feeding honey water (TPW-H-F) which contained fewer exclusive taxa (60) than that of females produced by feeding tetracycline hydrochloride (TPW-TH-F, 87 taxa) and females produced with high temperature (TPW-HT-F, 103 taxa). TPW-S-F and TPW-S-M shared the most taxa than with others. A core microbiome, i.e., taxa detected in all samples, was compiled for all the *T. pretiosum* sets, with nine common taxa detected. TPW-TH-F and TPW-HT-F had no unique OTUs.

Total bacterial alpha diversity was calculated between *T. pretiosum* microbiome taxonomic profiling sets, TPW-H-F, TPW-S-F, TPW-TH-F, TPW-HT-F, TPW-S-M, using metrics for species richness (Ace, Chao, and number of OTUs in each set), diversity index (Shannon and Simpson), and coverage rate (Table 2). Different *T. pretiosum* sets had their own alpha diversity characteristics. The Chao index of TPW-TH-F (*p* < 0.05), TPW-HT-F (*p* < 0.001), and TPW-S-M (*p* < 0.001) was significantly higher than that of the TPW-H-F. The Chao index of TPW-HT-F was 104.64, indicating that the microbial overall abundance and species diversity were high. Additionally, the Ace index of 106.40 was close to the Chao index, and the Chao index and Ace index were reliable for estimating the number of OTUs. The Ace index of TPW-HT-F (*p* < 0.001) and TPW-S-M (*p* < 0.001) was significantly higher than that of the TPW-H-F. TPW-S-M was ranked second in Chao and Ace indices, also indicating that the microbial overall abundance and species diversity were higher. The Shannon index of TPW-TH-F (*p* < 0.05) was significantly higher than that of the TPW-H-F. The Simpson index of TPW-S-F (*p* < 0.05) and TPW-TH-F (*p* < 0.001) was significantly lower than that of the TPW-H-F. The Chao index, Ace index, and Shannon index of TPW-H-F were the lowest, while the Simpson index of TPW-H-F was the largest, indicating that the microbial species’ overall abundance and diversity was low and individual. In general, the alpha diversity of TPW-H-F samples was lower than detected in other samples.

The heatmap clustering analysis was conducted to study the species composition and abundance changes of endosymbiont bacteria in *T. pretiosum* females (Figure 2 and Appendix A). A genus-level hierarchical cluster analysis indicates the high-temperature rearing and control clusters as the groups sharing the most endosymbiont bacteria community structure. The community structure of endosymbiont bacteria in *T. pretiosum* reared with sulfadiazine exhibited the most divergence compared with the other samples, both in species composition and abundance. *Wolbachia* was detected in all *T. pretiosum* samples, and the abundance of *Wolbachia* was lower in samples TPW-S and TPW-TH. Moreover, the genera *Acinetobacter*, *Faecalibacterium*, *Paracoccus*, *Prevotella*, *Pseudomonas*, *Ruminococcus*, and *Staphylococcus* were detected in all *T. pretiosum* samples, as well.

The relative abundance of the top ten genera of endosymbiont bacteria in *Trichogramma* samples are shown in Table 3 (Appendix A). The bacterial compositions of the samples obtained under control and high-temperature treatments of *Wolbachia* were dominated with average relative abundance of 99.01% and 96.43%, respectively. After treatment with sulfadiazine and tetracycline hydrochloride, the relative abundance of *Wolbachia* decreased to 38.49% and 20.12%. With the decrease of *Wolbachia* relative content (as low as 0.28 in male to 38.49 in female), the diversity and relative content of other endosymbiotic bacteria in *T. pretiosum* were increased. The growth of other endosymbiont bacteria was inhibited by the high content of *Wolbachia*. In addition, no other endosymbiont bacteria with reproductive regulation were detected in the *T. pretiosum*. The genus *Acinetobacte*, innately resistant to many classes of antibiotics, was detected in all samples, including with sulfadiazine and tetracycline hydrochloride treatments.

*Wolbachia* dominated in the *T. pretiosum* microbial community, followed by *Acinetobacter* in the sulfadiazine treatment. Compared with high temperature (the relative abundance is 96.93%), antibiotics treatment exhibited clearly reduced *Wolbachia* titer (38.49% and 20.12%). Although sulfadiazine completely changed the reproductive mode of *T. pretiosum*, the relative abundance of *Wolbachia* after sulfadiazine treatment was higher than that after tetracycline hydrochloride treatment. This question requires further study to find the answer. The microbial composition of *T. pretiosum* cannot be greatly changed by high temperature. The results showed that we could use antibiotics to reduce the titer of *Wolbachia* more effectively and change its microbial composition.

## 3. Discussion

Tetracycline hydrochloride and high temperature were capable of partial arrhenotokous in the *T. pretiosum*. Notably, sulfadiazine inverted the reproductive mode entirely with 100% male offspring, and stable bisexual after two generations. *Wolbachia* as the dominated endosymbiotic bacteria reduced the diversity and relative content of other endosymbiotic bacteria in *T. pretiosum*. With the decreasing of *Wolbachia* by antibiotics, the diversity of endosymbiotic bacteria was increasing.

The male proportion per generation of *T. pretiosum* increased gradually with the prolongation of treatment with high temperature or tetracycline hydrochloride. In previous research, no extreme 100% male offspring could be obtained even though after multiple generations exposure to antibiotics (tetracycline and rifampin) or high temperature [4,29,30]. In our current study, female *T. pretiosum* produced 100% male after feeding with sulfadiazine, which was different from previous curing experimental results [4,9,13,14]. Moreover, the cured male and cured female mating produced normal bisexual offspring (observed continuously up to 6 generations). Different antibiotic effects might have different mechanisms of action. Sulfadiazine and sulfamethoxazole worked by inhibiting the enzyme dihydropteroate synthetase [31,32], whereas rifampicin inhibited bacterial DNA-dependent RNA synthesis [33], tetracycline inhibited protein synthesis by preventing the association of aminoacyl tRNA with the bacterial ribosome [34]. In terms of inhibiting PI *Wolbachia*, sulfadiazine might outperform tetracycline and rifampicin.

According to the haplo-diploid sex determination system, haploid males develop from unfertilized eggs and diploid females develop from fertilized eggs. The proportion of male offspring less than 50% in F2(S) and F2(H) after reverting to bisexual populations should be explained by the fact that unfertilized eggs are commonly found in bisexual populations [35,36]. Offspring sex ratios in parasitoid wasps can be influenced by male precedence in emergence, maternal age, host size, and so on [37,38].

Though the reproductive mode of *T. pretiosum* was completely changed (from thelytoky to entirely arrhenotoky), *Wolbachia* could be detected by 16S rRNA-seq after sulfadiazine feeding. The male proportion in the secondary generation of *T. pretiosum* after sulfadiazine feeding was 100%. Similarly, after completion of development (from egg to the adult) in artificial host egg with tetracycline, *Wolbachia* inducing thelytokous *T. pretiosum* produced 100% male offspring in tested generations F2-F5 [10]. Furthermore, we discovered that males who reproduced by consuming sulfadiazine were capable of egg fertilization. Bisexual reproduction could be restored by crossing cured females. In populations with PI *Wolbachia* infection has developed into a fixed population, antibiotics are used to cure *Wolbachia*-infected females even if crossed individuals cannot produce female offspring [39,40,41,42]. Neither cured male nor the infected female were able to fertilize eggs after mating to produce female offspring, and the male were sterile [13,43]. In arthropods, the prevalence of *Wolbachia* varies within and among taxa, and no co-speciation events were noted [44].

Tetracycline hydrochloride reduced the parasitic egg number of the *T. pretiosum*, which was consistent with those reported by previous studies [9]. High temperature reduced the number of parasitic eggs of *Trichogramma* and the optimal temperature was different for different *Trichogramma* [42]. Additionally, the number of *T. pretiosum* parasitic eggs increased with the increase of high-temperature treatment generations. Compared with 25 °C, high temperature inhibited the parasitic eggs number of *T. pretiosum* F0 and F1 in the experiment, and the results were consistent with those reported by previous studies [45,46]. The emergence rate of *T. pretiosum* treated with high temperature or tetracycline hydrochloride gradually decreased with the increase of generations (high-temperature treatment for 11 generations, tetracycline hydrochloride treatment for six generations). However, sulfadiazine had a positive effect on the emergence rate of the *T. pretiosum*. The emergence rate of *T. pretiosum* increased after *Wolbachia* cure, which supports previous reports that the fecundity of *Trichogramma* infected with PI *Wolbachia* was much lower than that of uninfected individuals [47,48,49].

Most studies on insect–microbial interactions have focused on the obligate endosymbiotic relationships of one or several species, such as *Wolbachia* [50,51]. Notably, the antibiotics and high-temperature treatment changes microbiota for *Trichogramma* were generally ignored. *Wolbachia* infections reduced the diversity of microbials and changed the structure of microbials. The endosymbionts and other bacteria depend on limited resources and space to survive, and they might be in a competitive relationship [52,53], which would reduce the abundance of less competitive bacteria. Furthermore, endosymbionts can induce host immune responses (such as antibacterial peptides), which in turn may regulate bacterial diversity [54,55]. Here, the content of *Wolbachia* in the population was extremely high (up to 99%) in the thelytokous *T. pretiosum* induced by PI *Wolbachia*. It was shown that *Wolbachia* manipulates bacterial diversity in its host [56], as it achieved the degree fixed in the host population [57]. After treatment with antibiotics and high temperatures, the relative abundance of *Wolbachia* decreased to 20.12%, whereas the decrease in *Wolbachia* promoted the increase in co-bacterial diversity in *Trichogramma*. In the case of the brown planthopper, the *Wolbachia*-free group possessed higher microbial diversity than the *Wolbachia*-infected group [58]. Inconsistently, tetracycline decreased fly bacterial diversity and induced modifications in both bacterial abundance and relative frequencies [59]. With the contents of *Wolbachia* dropping, the proportion of males grew. The result was supported by results in the close-relationship species, *T. dendrolimi*, in which decreased *Wolbachia* titer caused a gradual change in the masculinization of intersex individuals [18].

Using 16S rRNA-seq, a core microflora was identified in the *T. pretiosum* population, indicating that the microflora of the population represents an important library of biodiversity. In addition to comparing the microbial composition of *T. pretiosum* population with different treatments, this study also provides evidence for the *Wolbachia* lineage. Microbiome analysis found that *Wolbachia* and *Acinetobacter* were the major core members of the microbiome among these treatments. Though *Acinetobacter* is not capable of enhancing the biotic fitness of the host [60], it is abundant in mosquito [61] and in arrhenotokous parasitoid, *Diglyphus wani* [62]. The core member of *Acinetobacter* should be the result of the high rates of multidrug resistance and extensive drug resistance [63]. Through antibiotics treatment, the changes of the microbial community in *T. pretiosum* were different, which might be due to the different resistances of the microbial community to different antibiotics [64]. After sulfadiazine treatment, the microbial community of females and males were similar. Tetracyclines and sulfadiazine are broad-spectrum antibiotics that were widely used in Gram-positive and Gram-negative bacteria [65,66].

Similar to our results, *Wolbachia* in insects or nematodes failed to be completely eliminated by antibiotics or high-temperature treatment [6,67]. It was worth noting that sulfadiazine caused PI *Wolbachia*-infected *T. pretiosum* to restore arrhenotoky parthenogenesis and bisexual reproduction, at least successive arrhenotoky or bisexual reproduction for six generations, despite rearing with honey water. Thereby, although the infection was not totally cured, *Wolbachia* titer was decreased to restore 100% arrhenotokous.

## 4. Materials and Methods

### 4.1. Insects

The thelytokous *T. pretiosum*, was originally bought from ARBICO Organics, Arizona, USA. The identification of the species was confirmed by *ITS2* [68] sequencing (Accession Number: MH890848) and comparing the DNA sequence with genome of *T. pretiosum* (Taxonomy ID: 7493). All wasps were maintained and reared on *Wolbachia* free eggs of *Corcyra Cephalonica* reared on flour containing 0.25% tetracycline hydrochloride (98% of certified purity; CAS: 64-75-5; Amresco, Solon, OH, USA). Eggs of *C. cephalonica* were killed by UV irradiation for 50 min and glued onto cards with 200 eggs per sheet. Wasps were fed on 30% honey water (acacia honey mixed with distilled deionized water in a ratio of 3:7) and reared at 25 ± 1 °C, 70 ± 5% relative humidity (RH), and a 14:10 (L:D) h photoperiod in the laboratory.

### 4.2. Antibiotics and High-Temperature Treatment

Thelytokous *T. pretiosum* were treated with either antibiotics curing (5.0 mg/mL sulfadiazine, 10.0 mg/mL tetracycline hydrochloride) or high-temperature treatment (30 ± 1 °C) [4]. Both antibiotics were mixed with 30% honey. *T. pretiosum* reared at a temperature of 25 ± 1 °C was used as the control. Every newly emerged female was allowed to oviposit for 24 h. The number of parasitic eggs, emergence rate, and male percentage of offspring were recorded for 5 generations by checking under a dissecting microscope Leica S8AP0 (Leica Microsystems, Nussloch GmbH, Wetzlar, Germany). Ten replicates for each treatment and control were detected.

### 4.3. Crossing Experiments

To investigate the fertility of *T. pretiosum* after treatment with sulfadiazine, we employed cross-generational treatment, as shown in Figure 3. Both sexes were fed with sulfadiazine or honey water for 6 generations. In the first generation, females (F0) were reared with sulfadiazine. Then, 100% male appeared in the offspring of generation 2 (F2), which was backcrossed with F1 female to obtain bisexual (F3). Alternatively, females of F1 and F3 were reared with honey water or with sulfadiazine to evaluate whether females reverted to arrhenotoky without sulfadiazine.

### 4.4. Wolbachia Infection Detection

The genomic DNA extractions were performed using the trace-DNA templates method [69], which extracted DNA from individual wasps. DNA extraction buffer was mixed with protease K (20 mg/mL) and STE buffer (100 mmol/L NaCl, 10 mmol/L Tris-HCl, 1 mmol/L EDTA, pH = 8.0) in a ratio of 1:19. Wasps were crushed individually with 1 µL DNA extraction buffer in sterile Eppendorf tubes with the tip of a sterile pipette. Using another 20 µL of DNA extraction buffer to wash the pipette and immerse tissues, samples were maintained at 56 °C for 4 h, 95 °C for 10 min. Finally, the supernatant was used as the DNA template. Primers for *ITS2* gene were used to test the quality of extracted DNA [68]. *Wolbachia* infection was detected by the *wsp* gene’s specific primers [70]. Amplification from *wsp* was sequenced (Tsingke Biotechnology Co., Ltd., Beijing, China) and analyzed using BLAST in the NCBI database. Ten wasps were detected according to the frequency of *Wolbachia* infection in each treatment or generation.

### 4.5. 16S rRNA-Seq Sample Collection

According to the *Wolbachia* infection detected, five groups of *T. pretiosum* from control, sulfadiazine (female F1 and male F2), tetracycline hydrochloride (F5), high temperature (F5), were performed 16S rRNA-seq. About 250 wasps per sample were collected and quickly frozen in liquid nitrogen, stored at −80 °C for further 16S rRNA-seq by Beijing Genomics institution (Beijing, China).

### 4.6. Statistics

Significant differences of the number of parasitic eggs, emergence rate and male percentage of offspring, alpha diversity, or relative abundance of bacterial communities between treatments and control were evaluated by *t*-tests using SPSS 24.0 software (IBM SPSS Statistics, IBM Corporation, Chicago, IL, USA). All analyses were considered significant at *p* < 0.05. The statistical calculation of each parameter was performed as follows. The number of parasitic eggs: the number of black *C. cephalonica* eggs on the egg card was recorded as the egg parasitism number of single female on days 7–8 of egg card culture (counted with dissecting microscope). Emergence rate: the number of wasps/the number of parasitic eggs × 100%. Male percentage: the number of males/the number of emergence wasps × 100%. Infection rate: the number of positive wasps/the number of detected wasps × 100%.

Raw reads of 16S rRNA-seq were filtered to remove adaptors and low-quality and ambiguous bases, and then paired-end reads were added to tags by the Fast Length Adjustment of Short reads program (FLASH, v1.2.11) [71] to obtain the tags. The tags were clustered into OTUs with a cut-off value of 97% using UPARSE software (v7.0.1090) [72] and chimera sequences were compared with the GOLD database using UCHIME (v4.2.40) [73] to detect. Then, OTU representative sequences were taxonomically classified using Ribosomal Database Project (RDP) Classifier v.2.2 with a minimum confidence threshold of 0.6, and trained on the green genes Database v201305 by QIIME v1.8.0 [74]. The USEARCH_global [75] was used to compare all tags back to OTU to obtain the OTU abundance statistics table of each sample. Alpha and beta diversity were estimated by MOTHUR (v1.31.2) [76] and QIIME (v1.8.0) [74] at the OTU level, respectively. A sample cluster was conducted by QIIME (v1.8.0) [74] based on UPGMA. KEGG and COG functions were predicted using the PICRUSt software [77].

## 5. Conclusions

In the current study, although sulfadiazine did not eliminate *Wolbachia*, it was found to be more effective in restoring entirely arrhenotokous and successive bisexual reproduction in infected *T. pretiosum*. The male produced by *T. pretiosum* feeding on sulfadiazine had reproductive function. Antibiotics reduced the titer of *Wolbachia* and increased the diversity of endosymbiotic bacteria, which resulted in switching the reproductive modes between thelytoky and arrhenotoky.

## Figures and Tables

**Figure 1 ijms-24-08448-f001:**
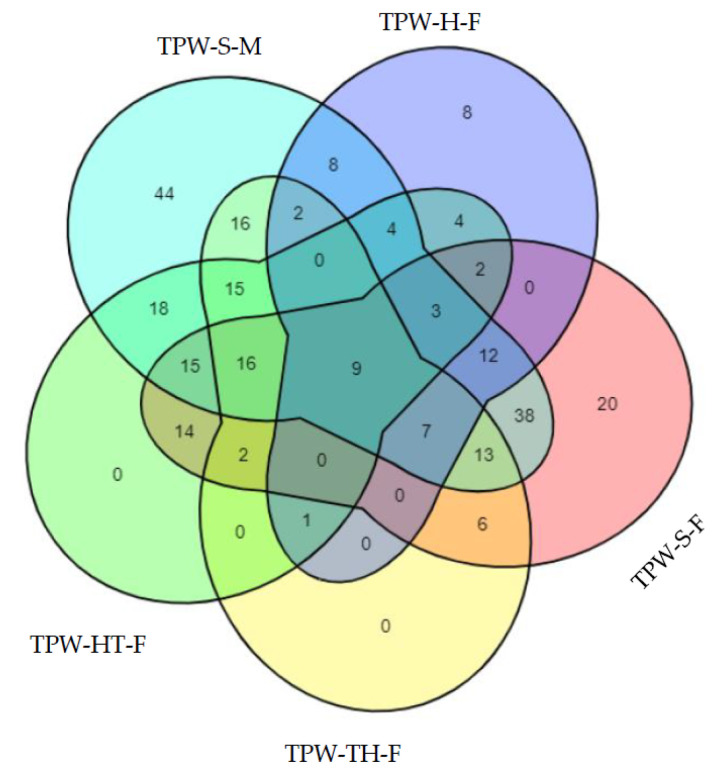
Venn diagram of bacterial communities showing the number of shared and exclusive bacterial taxa is shown relative to *T. pretiosum*. TPW-H-F: females produced by females fed honey water; TPW-S-F: females produced by females fed sulfadiazine, TPW-TH-F: females produced by females fed tetracycline hydrochloride, TPW-HT-F: females produced by females with high temperature, TPW-S-M: males produced by females fed sulfadiazine. The table and figure abbreviations hereinafter are the same as in Figure 1.

**Figure 2 ijms-24-08448-f002:**
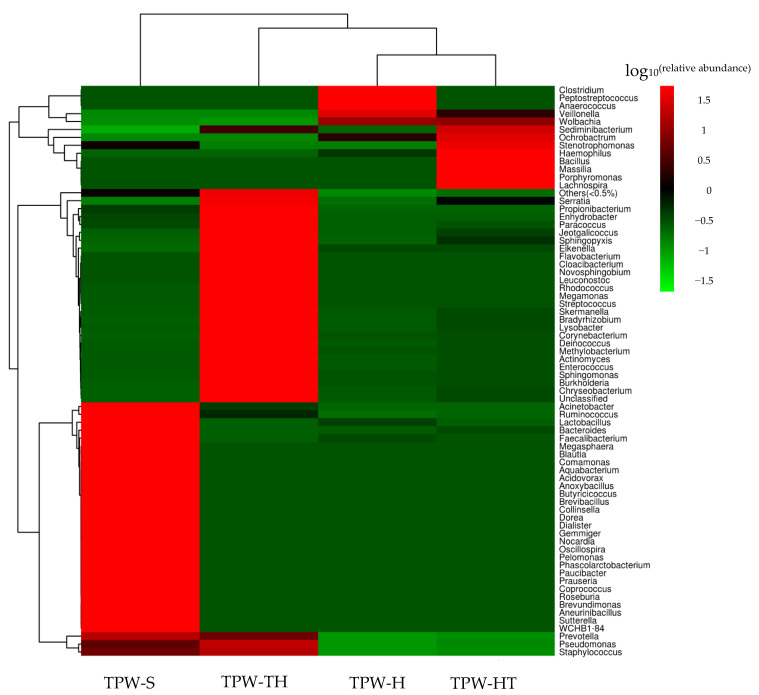
Heatmap showing log_10_^(relative abundance)^ of *T. pretiosum* females’ abundant bacterial genera.

**Figure 3 ijms-24-08448-f003:**
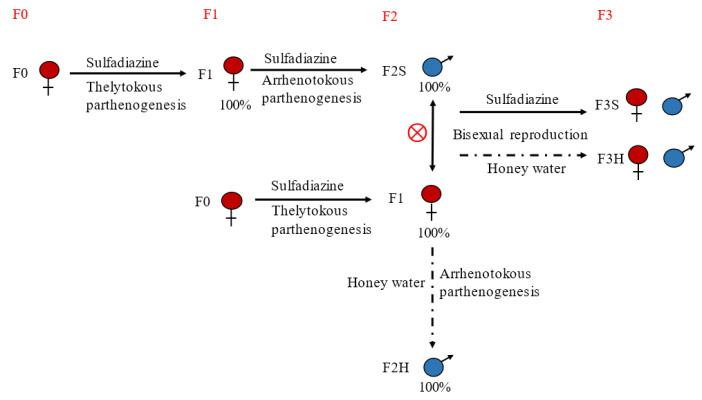
Illustration of crossing design for *T. pretiosum*. Full arrows indicate sulfadiazine treatment. Dotted arrows indicate honey water treatment. 
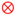
 The red symbol with a circle indicates male-female hybridization. F2S was generated after F1 was fed with sulfadiazine. F2H was generated by F1 fed with 30% honey water. F3S: the F2 male hybridized with the F1 female and fed 5.0 mg/mL sulfadiazine to produce F3S. F3H: the F2 male hybridized with the F1 female and fed 30% honey water to produce F3H.

**Table 1 ijms-24-08448-t001:** Effects of three treatments on biological characteristics of *T. pretiosum*.

Treatments	Generations	The Average Number of Parasitic Eggs	The Average Emergence Rate (%)	The Average Male Percentage (%)	*Wolbachia* Infection (*n* = 10)	N
Control	F0	29.40 ± 0.83	94.37 ± 1.12	0.00 ± 0.00	100%	294
Sulfadiazine	F0	26.90 ± 1.10	105.39 ± 2.31 **	0.00 ± 0.00	10%	269
	F1(S)	29.30 ± 1.38	101.83 ± 1.78 *	100.00 ± 0.00 **	0%	293
	F1(H)	29.60 ± 1.17	97.29 ± 1.15 *	100.00 ± 0.00 **	0%	296
	F2(S) bisexual	28.60 ± 0.52	105.25 ± 3.51 **	29.66 ± 7.04 **	0%	286
	F2(H) bisexual	29.90 ± 0.50	104.14 ± 3.85 *	29.20 ± 7.55 **	0%	299
Tetracycline Hydrochloride	F0	14.40 ± 1.43 **	84.08 ± 5.86 *	6.87 ± 2.76 *	100%	144
	F1	20.30 ± 0.93 **	82.29 ± 2.48 **	15.24 ± 3.02 **	100%	203
	F2	24.30 ± 1.80 *	80.52 ± 3.28 **	17.31 ± 2.23 **	100%	243
	F3	21.80 ± 1.67 **	85.87 ± 2.53 *	14.92 ± 3.89 **	100%	218
	F4	23.70 ± 2.30 *	84.05 ± 2.99 *	28.14 ± 10.53 **	100%	237
High Temperature	F0	13.20 ± 0.93 **	97.59 ± 1.24	0.71 ± 0.71	100%	132
	F1	28.40 ± 0.50	95.33 ± 1.64	2.54 ± 2.16 *	100%	284
	F2	34.60 ± 0.58 **	89.90 ± 1.82	2.35 ± 1.24 *	100%	346
	F3	33.70 ± 0.50 **	85.00 ± 2.80 *	8.20 ± 1.96 **	100%	337
	F4	36.70 ± 0.54 **	84.83 ± 2.17 *	9.22 ± 2.20 **	100%	367

The values are the mean ± standard error. “*” and “**” indicate significant differences at *p* < 0.05 and *p* < 0.001, respectively, between treatment and control. “S” indicates sulfadiazine treatment; “H” indicates honey water treatment. “N” represents the total numbers of observed offspring.

**Table 2 ijms-24-08448-t002:** Alpha diversity measures of bacterial communities in *T. pretiosum*.

Sample	Diversity Index					Coverage Rate (%)
	OTUs	Chao	Ace	Shannon	Simpson	
TPW-H-F	60	35.42 ± 2.79	49.17 ± 9.50	0.27 ± 0.078	0.92 ± 0.023	100.00 ± 0.00
TPW-S-F	157	77.00 ± 24.48	80.05 ± 23.21	1.84 ± 0.40	0.32 ± 0.060 *	100.00 ± 0.00
TPW-TH-F	87	87.00 ± 17.59 *	87.00 ± 10.60	3.41 ± 0.54 *	0.07 ± 0.016 **	100.00 ± 0.00
TPW-HT-F	103	104.64 ± 7.80 **	106.40 ± 13.25 **	0.29 ± 0.11	0.93 ± 0.072	99.96 ± 0.017
TPW-S-M	220	101.33 ± 14.19 **	101.07 ± 15.68 **	2.12 ± 0.51	0.4 ± 0.14	100.00 ± 0.00

The values are the mean ± standard error. “*” and “**” indicate significant differences at *p* < 0.05 and *p* < 0.001, respectively, between treatment and control (TPW-H-F).

**Table 3 ijms-24-08448-t003:** Relative abundance (%) of the top ten genera of endosymbiont bacteria in *T. pretiosum* samples by 16S rRNA-seq.

TPW-H-F	TPW-S-F	TPW-TH-F	TPW-HT-F	TPW-S-M
*Wolbachia* 99.01 ± 2.30	*Wolbachia* 38.49 ± 14.44 **	*Wolbachia* 20.12 ± 2.31 **	*Wolbachia* 96.43 ± 1.48	*Wolbachia* 0.28 ± 0.27 **
*Acinetobacter* 0.72 ± 0.31	*Acinetobacter* 40.59 ± 6.27 **	Unclassified 13.06 ± 4.01	Unclassified 0.59 ± 0.31	*Acinetobacter* 63.25 ± 17.99 **
*Pseudomonas* 0.04 ± 0.06	*Pseudomonas* 1.22 ± 0.28	*Enhydrobacter* 12.16 ± 3.49	*Acinetobacter* 0.39 ± 0.16	*Pseudomonas* 2.95 ± 1.72
*Pelomonas* 0.02 ± 0.018	*Pelomonas* 1.56 ± 0.88	*Burkholderia* 7.60 ± 1.52	*Burkholderia* 0.33 ± 0.09	*Pelomonas* 1.86 ± 0.64
*Prevotella* 0.18 ± 0.31	*Prevotella* 1.23 ± 1.68	*Bradyrhizobium* 5.78 ± 1.94	Others (<0.5%) 0.30 ± 0.06	*Prevotella* 1.46 ± 2.35
*Megasphaera* -	*Megasphaera* 1.85 ± 3.21	*Acinetobacter* 4.90 ± 1.11 **	*Bradyrhizobium* 0.30 ± 0.07	*Megasphaera* 0.75 ± 1.29
*Enhydrobacter* 0.03 ± 0.05	*Enhydrobacter* 0.63 ± 0.24	*Sphingomonas* 4.14 ± 1.08	*Serratia* 0.27 ± 0.11	*Enhydrobacter* 1.88 ± 0.80
*Coprococcus* -	*Coprococcus* 0.76 ± 0.85	Others (<0.5%) 3.78 ± 1.07	*Sphingomonas* 0.14 ± 0.02	*Coprococcus* 1.50 ± 1.73
*Ruminococcus* -	*Ruminococcus* 0.38 ± 0.62	*Methylobacterium* 3.61 ± 1.01	*Prevotella* 0.12 ± 0.06	*Ruminococcus* 1.75 ± 2.92
*Butyricicoccus* -	*Butyricicoccus* 0.81 ± 1.29	*Prevotella* 2.51 ± 0.61	*Staphylococcus* 0.10 ± 0.009	*Butyricicoccus* 1.27 ± 2.19

“-” indicates no detection of endosymbiont in this genus. The values are the mean ± standard error. “**” indicates significant differences at *p* < 0.001, between treatment and control (TPW-H-F).

## Data Availability

The data presented in this study are available on reasonable request from the first author.

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
