# Peer review of "Bacterial Community Survey of Wolbachia-Infected Parthenogenetic Parasitoid Trichogramma pretiosum (Hymenoptera: Trichogrammatidae) Treated with Antibiotics and High Temperature"

_ijms, 2023, doi:10.3390/ijms24098448_

Round 1

Reviewer 1 Report (Previous Reviewer 2)

The manuscript was deeply revised to remove the data and results of T. cacoeciae and T. embryophagum which caused thelytoky by heredity. And  discuss in detail the changes of bacterial community after restoring bisexual reproduction in the T. pretiosum which induced thelytoky by Wolbachia. By using antibiotics and high temperature, the elimination of Wolbachia was studied. This makes the purpose and design of the study clearer and removes the ambiguity in the previous version, because the type of thelytoky caused by heredity is not a control to compare.

So far, other studies have shown that Wolbachia can be completely removed. In this paper, quantitative PCR and high-throughput 16S ribosomal RNA sequencing are used to detect that Wolbachia is not completely eliminated, but once the titer of Wolbachia is lower than the threshold that thelytoky requires, thelytoky will revert to arrhenotoky.

The manuscript has only a few minor errors, such as the absence of italics for Trichogramma in the title.

Author Response

Dear reviewers:

Thank you for your comments concerning our manuscript. Those comments are all valuable and very helpful for revising and improving our paper, as well as the important guiding significance to our researches. All changes (marked in red) according to the reviewers’ advice will not influence the content or framework of the paper. We appreciate the editors and reviewers’ warm work and earnestly hope that the correction will meet with approval. Our answers to the reviewer’ s commentsare as followed:

1. Response to comment: The manuscript has only a few minor errors, such as the absence of italics for Trichogramma in the title.

We thank you for the reminder. I have made italic changes to the names of species in this article and revised the full text.

Reviewer 2 Report (New Reviewer)

The manuscript described the presence of the bacteria in the Wolbachia-infected partheno-genetic parasitoid Trichogramma pretiosum.

I read the manuscript with interest and consider that it could have potential interest for the communities of scientists who raised the T pretiosum for biological control.

However, I need to address a few points that can be improved:

1) In abstract line 20-21, "Although antibiotics did not eliminate Wolbachia in T. pretiosum...." contradict Table 1, where the Wolbachia infection was shown as 0%. It will help if they explain the Wolbachia infection title under the table better.

In Introduction,

line 75, they could give a sentence or two about the advantage and benefits of bacterial community study in T. pretiosum.

Results

line 94 explains the "reverted"

Table 1:

The number of Parasitic Eggs should be whole, like 29+ 1, as 0.4 eggs can't produce anything

Line 210 with the degreasing do you mean decreasing?

In Material and methods

Line 325 You need to explain 30% honey water. Is it honey or 30% of sucrose solution? If it is honey, then honey itself can have a different composition, then you need to add what kind of honey you use.

 Line 330 Antibiotic treatment: 5.0mg/mL is the solution of the treatment, like 5.0mg per mL of 30% honey water. Or you add this treatment to 30% honey water... and then the final concentration of the feeding will be less. Can you clarify?

Author Response

Dear reviewers:

Thank you for your comments concerning our manuscript. Those comments are all valuable and very helpful for revising and improving our paper, as well as the important guiding significance to our researches. All changes (marked in red) according to the reviewers’ advice will not influence the content or framework of the paper. We appreciate the editors and reviewers’ warm work and earnestly hope that the correction will meet with approval. Our answers to the reviewer’ s comments point by point are as followed:

1. Response to comment: In abstract line 20-21, "Although antibiotics did not eliminate Wolbachia in T. pretiosum...." contradict Table 1, where the Wolbachia infection was shown as 0%. It will help if they explain the Wolbachia infection title under the table better.

Wolbachia can be detected by 16S rRNA-seq but can not be dected by normal PCR in the table 1. The reseasons for the different outputs from the two methods used to detect Wolbachia should be due to sampes and the sensitity. Considering the two detection methods and results, it can be concluded that sulfadiazine did not eliminate Wolbachia in T. pretiosum.

Table 1 Comparing two methods used to detected Wolbachia in the manuscript

Methods

Sample usded to DNA extraction

Sensitity

Wolbachia

Normal PCR

One wasp

Lower sensitive

-

16S rRNA-seq

Pool sample about 250 wasps

Higher sensitive

+

2. Response to comment: In Introduction, line 75, they could give a sentence or two about the advantage and benefits of bacterial community study in T. pretiosum.

Sentence in the introduction (line 81-85) as followed. Compared with other experimental methods, 16S rRNA-seq can simultaneously assess microbial diversity and functional activity potential of each taxon. The research on the bacterial community of T. pretiosum will help us understand the core microflora and their potential capacity to provide responses to the environmental changes through the production modes in T. pretiosum.

3.Response to comment: Results line 94 explains the "reverted".

"Reverted": Reverted line is the first-generation feed on sulfadiazine, in which the production mode of female T. pretiosum was reverted from thelytoky to complete / absolute arrhenotoky. To avoid the confusion, we used “the reverted line females (F1)” in the version.

4. Response to comment: Table 1: The number of Parasitic Eggs should be whole, like 29+1, as 0.4 eggs can't produce anything.

The number of parasitic eggs is a natural number in the actual experiment. The number of parasitic eggs was changed to the Average Number of Parasitic Eggs in Table 1. According to the statistical analysis, 29.40 ± 0.83 is the mean ± standard error (SE). For example, the fertility rates (births per woman) in China is 1.3 according to the United Nations Population Division reported. ( https://data.worldbank.org/indicator/SP.DYN.TFRT.IN)

5. Response to comment: Line 210 with the degreasing do you mean decreasing?

Thank you for the carefully review. It has been revised in the manuscript.

6. Response to comment: In Material and methods: Line 325 You need to explain 30% honey water. Is it honey or 30% of sucrose solution? If it is honey, then honey itself can have a different composition, then you need to add what kind of honey you use.

30% honey water is solution mixed with acacia honey (produced by Jesitte, Shenzhen, China) and distilled deionized water in a ratio of 3: 7.

7. Response to comment:  Line 330 Antibiotic treatment: 5.0 mg/mL is the solution of the treatment, like 5.0 mg per mL of 30% honey water. Or you add this treatment to 30% honey water... and then the final concentration of the feeding will be less. Can you clarify?

1.0 mL of 5.0 mg/mL sulfadiazine solution is constituted by 1.0 mL of 30% honey water (solvent) and 5.0 mg of sulfadiazine powder (solute). We explained it in detail in the materials and methods.

Reviewer 3 Report (New Reviewer)

This MS entitled “Bacterial Community Survey of Wolbachia-infected Parthenogenetic 

Parasitoid Trichogramma pretiosum (Hymenoptera: Trichogrammatidae) Treated 

with Antibiotics and High Temperature” by Gu and colleagues describes that the effects of antibiotic treatments and high temperature treatment on Wolbachia infection levels and parthenogenetic conditions. Additionally, the microbial state changes of host insects after these treatments were measured by 16S rRNA-sequence. However, it is not clear why the states of microorganisms were compared, and the significance of this analysis in this MS is not fully documented. The connection between the first half of the experiment (antibiotic treatments) and the second half of the experiment (16S rRNA seq) is not sufficiently explained. The conclusion of this MS is not clear.

Author Response

Dear reviewers:

Thank you for your comments concerning our manuscript. Those comments are all valuable and helpful for revising and improving our paper, as well as the important guiding significance to our research. All changes (marked in red) according to the reviewers’ advice will not influence the content or framework of the paper. We appreciate the editor's and reviewers’ warm work. Our answers to the reviewer's comments point by point are as follows:

Response to comment: (1) However, it is not clear why the states of microorganisms were compared, and the significance of this analysis in this MS is not fully documented. (2) The connection between the first half of the experiment (antibiotic treatments) and the second half of the experiment (16S rRNA-seq) is not sufficiently explained. (3) The conclusion of this MS is not clear.

>(1) Wolbachia are a bacterial endosymbiont present in most insect species (Warecki et al., 2022). As most reported, Wolbachia play an important role in the regulate their host’s reproduction, including cytoplasmic incompatibility (Breeuwer and Werren, 1990), Parthenogenesis indcution (Tortora et al., 2007), male killing (Hackett et al., 1986), and Feminization (Fujii et al., 2001). The reproductive mode of T. pretiosum infecting with parthenogenesis-inducing (PI) Wolbachia will be changed to varying degrees by exposure to antibiotics and high temperatures (Stouthamer et al., 1990). As we showed, the T. pretiosum exposed to sulfadiazine, tetracycline treatment, and high temperature produced about 100%, 28%, and 9.22% male offspring, respectively (Table 1). We hypothesize that the bacterial community (diversity and relative abundance) and the core microflora have different potential capacities to respond to the antibiotics and high temperature leading to the changing of reproductive modes in T. pretiosum. Reducing the titer of Wolbachia, the higher diversity of the bacterial community drives a reproductive mode switch in T. pretiosum between thelytoky and arrhenotoky. This sentence is also explained in the manuscript.

>(2) The link between antibiotic treatments and 16S rRNA-seq has been additionally explained in the introduction (Line 66-70).

 The reproductive mode of T. pretiosum infecting with PI Wolbachia will be changed to varying degrees by exposure to antibiotics and high temperatures [4,16]. We hypothesize that the bacterial community and the core microflora have different potential capacities to respond to the antibiotics and high temperature leading to the changing of reproductive modes in T. pretiosum.

>(3) The conclusion has been supplemented in the article to make it more clearly.

>The effect of antibiotics on T. pretiosum and the endosymbiotic bacteria changed before and after antibiotics were administered, according to the conclusion in the correction.

 In the current study, though sulfadiazine did not eliminate Wolbachia, it was found to be more effective in restoring entirely arrhenotokous and successive bisexual reproduction in infected T. pretiosum. The male produced by T. pretiosum feed on sulfadiazine had reproductive function. Antibiotics reduced the titer of Wolbachia and increased the diversity of endosymbiotic bacteria, which resulted in switching the reproductive modes between thelytoky and arrhenotoky.

Yours sincerely.

Corresponding author: Xiaofang He

E-mail: hexf@scau.edu.cn

Round 2

Reviewer 3 Report (New Reviewer)

This MS entitled “Bacterial Community Survey of Wolbachia-infected Parthenogenetic Parasitoid Trichogramma pretiosum(Hymenoptera: Trichogrammatidae) Treated with Antibiotics and High Temperature” by Guo and colleagues describe the effect of antibiotic treatments and heat stress on wolbachia infection rates. Wolbachia infection alters the sex rate of offspring which generated by parthenogenesis. Sulfadiazine, one of antibiotic used in the MS, recovered completely the sex rate of offspring born from parthenogenesis. The authors analyzed microbiota changes after antibiotic treatments and heat treatment. The microbiota was different between treatments. These might reflect the effect of treatments on reproductivity. The data were clearly showed but a part of the statements should be reconsidered.

Major concern,

 The author concluded that the threshold of wolbachia titer might regulate the sex rate of offspring of parthenogenesis. Absolute quantification of Wolbachia was not mentioned in the MS, only relative data were put. If the authors discuss the threshold, they should include quantitative results in the MS. 

Minor concerns,

1. Why the sex ratio of F2(S) and F2(H) were lower than 50%? Please discuss in the MS.

2. Line 216-217, Did you do statistical analysis to table 3?

3. In discussion section, the paragraph (line 227-242) was not based on the data of this MS. Please remove or rewrite.

4. Line 272, “Antibiotics and high temperature on fecundity of Trichogmramma spp.” should be change to bold. 

Author Response

Dear Editors and Reviewer:

Thank you for your letter and for the reviewers’ comments concerning our manuscript (ID: ijms-2330612). We appreciate the time and effort that you and the reviewers have dedicated to providing your valuable feedback on our manuscript. We have been able to incorporate changes to reflect most of the suggestions provided by the reviewers. The changes were highlighted in red within the manuscript.

# Reviewer3:

Thank for your valuable advice. We have revised all of them as follows:

Major concern

Comment 1: The author concluded that the threshold of Wolbachia titer might regulate the sex rate of offspring of parthenogenesis. Absolute quantification of Wolbachia was not mentioned in the MS, only relative data were put. If the authors discuss threshold, they should include quantitative results in the MS. 

Response: It is great advice. We deleted all the sentences about threshold in the revised version throughout the manuscript, abstract, and discussion sections.

The threshold should be a very important question answered to the introduction modified by Wolbachia. We will explore it in the future.

Minor concerns

Comment 1: Why the sex ratio of F2(S) and F2(H) were lower than 50%? Please discuss in the MS.

Response:Thank you for pointing this out. We have rewritten this part and made it further clarified as follow: According to the haplo-diploid sex determination system, haploid males develop from unfertilized eggs, and diploid females develop from fertilized eggs. The proportion of male offspring less than 50% in F2(S) and F2(H) after reverting to bisexual populations should be explained by the fact that unfertilized eggs are commonly found in bisexual populations [Oliveira et al., 2003; Lü et al, 2017]. Offspring sex ratios in parasitoid wasps can be influenced by male precedence in emergence, maternal age, host size, and so on [King 1987; Hiehata and Suzuki 1985].

Comment 2: Line 216-217, Did you do statistical analysis to table 3?

Response: We are grateful to the reviewer for the insightful comments on the data. We have performed a statistical analysis of the data in Table 3 and explained the process in the section on material and method.

Comment 3: In discussion section, the paragraph (line 227-242) was not based on the data of this MS. Please remove or rewrite.

Response: Thank you for pointing this out. We agree with this comment. Therefore, we have removed the line 227-242 from this article.

Comment 3: Line 272, “Antibiotics and high temperature on fecundity of Trichogmramma spp.” should be change to bold. 

Response: Agree. I've modified as “Effects of antibiotics and high temperature on fecundity.” in bold.

Yours sincerely.

Corresponding author: Xiaofang He

E-mail: hexf@scau.edu.cn

Round 3

Reviewer 3 Report (New Reviewer)

In the revised manuscript, authors have addressed all of my concerns by clarification. The revised MS, however, still contains minor grammatical errors which should be corrected prior to publication. This MS contains sentences which are difficult to understand. The MS should be edited in English by a native speaker.

The sentences in line 53-54 and 290-291 may be incompleted.

Author Response

Dear Editors and Reviewer:

Thank you for your letter and for the reviewers’ comments concerning our manuscript (ID: ijms-2330612). We checked the grammar of the original manuscript and corrected some minor grammatical mistakes. Moreover, the paper has been carefully revised by Dr. Muhammad Musa Khan (https://loop.frontiersin.org/people/1182210/overview) to improve the grammar and readability. All the changes were highlighted in red within the manuscript.

Reviewer #3:

Your careful reminders have improved the quality of our manuscript.

Comment 1: In the revised manuscript, authors have addressed all of my concerns by clarification. The revised MS, however, still contains minor grammatical errors which should be corrected prior to publication. This MS contains sentences which are difficult to understand. The MS should be edited in English by a native speaker. The sentences in line 53-54 and 290-291 may be incomplete.

Response: Based on your advice, we have revised all of them as follows:

Line 53-54: Reducing the Wolbachia titer increases the proportion of males in the offspring. However, most antibiotics, including tetracycline hydrochloride, increased the proportion of male offspring and failed to eliminate Wolbachia infection.

Line 290-291: With the contents of Wolbachia dropping, the higher proportion of males.

Thank you again for your comments.

Yours sincerely.

Corresponding author: Xiaofang He

E-mail: hexf@scau.edu.cn

This manuscript is a resubmission of an earlier submission. The following is a list of the peer review reports and author responses from that submission.

Round 1

Reviewer 1 Report

This paper describes a series of experiments to determine the relationship of thelytoky induction in Trichogramma wasps by Wolbachia or by genetic factors. I am afraid I do not see much contribution of this paper to our understanding of thelytoky in Trichogramma. Almost all of the work that is presented here has already been done in the past. The authors show that two antibiotics work in permanently reverting thelytoky to arrhenotoky in Wolbachia infected Trichogramma, but are not effective in doing so in non-infected thelytokous Trichogramma (T. cacoeciae and T.embryophagum). This was already shown in the first few publications on this subject in Stouthamer et al, 1990) and Stouthamer and Werren, 1993.

In addition, they seem to think that the Wolbachia infection appears to linger in Trichogramma lines when treated with tetracycline, they even cite several papers on nematodes where this appears to be the case, that despite treatment with tetracycline the infection cannot be completely cured. This is no surprise because for nematodes the wolbachia may be an obligate symbiont. This is not the case in Trichogramma, Tetracycline can completely cure the wasps of their infection.

In addition, they show that the Trichogramma they rear on wolbachia free hosts still contain quite a community of bacteria, however the meaning of this is unclear because no control experiment was done by simply extracting the dna of the host eggs themselves. The bacteria found in the samples of the wasp may simply reflect the bacteria found in or on the eggs of the Corcyra. While the authors collect a lot of data from these microbiomes, what are the conclusions? And why does it matter?

The origin and identity of the Trichogramma used in the experiments are also unclear. When were these lines originally collected? The authors state their identity was confirmed using the sequence of ITS2 and blasting the sequences against genbank. There is no accession number given for the sequences they obtained, in GenBank there quite a large number of sequences deposited with the wrong species name given to them. In addition the literature is quite unclear of the difference between T. embryophagum and T. cacoeciae (see Sumer F, Tuncbilek AS, Oztemiz S, Rugman-Jones PF, & Stouthamer R (2009) A molecular key to the common species of Trichogramma of the Mediterranean region. Biocontrol 54(5):617-624.)

 .

Author Response

Dear Revivers,

We are thanks for the valuable suggestions in the referee’s report. We have made careful modifications on the manuscript. The point-by-point response was uploading with the manuscript.

Best Regards!

Yours sincerely,

Xiaofang

Response to comments

1.This paper describes a series of experiments to determine the relationship of thelytoky induction in Trichogramma wasps by Wolbachia or by genetic factors. I am afraid I do not see much contribution of this paper to our understanding of thelytoky in Trichogramma. Almost all of the work that is presented here has already been done in the past. The authors show that two antibiotics work in permanently reverting thelytoky to arrhenotoky in Wolbachia infected Trichogramma, but are not effective in doing so in non-infected thelytokous Trichogramma (T. cacoeciae and T. embryophagum). This was already shown in the first few publications on this subject in Stouthamer et al, 1990) and Stouthamer and Werren, 1993.

> Sure, the creative work about antibiotics reverting thelytokous reproduction to biosex had been done a few dozen years ago. Here, this study expands our understanding of how Wolbachia regulate host reproduction as a density-dependent model. Once the titer of Wolbachia is lower than the threshold that thelytoky requires, the thelytoky will revert to the arrhenotoky, such as weevil Pantomorus postfasciatus (Rodriguero et al., 2020). The result is supported by the close relationship species, T. dendrolimi, in which decreased Wolbachia titers cause a gradual change in the masculinization of intersex individuals (Zhang et al., 2022). It does not need to eliminate Wolbachia completely to cause the change in reproduction mode. The conclusion provides insight into the complexity of the Wolbachia and host.

  1. In addition, they seem to think that the Wolbachia infection appears to linger in Trichogramma lines when treated with tetracycline, they even cite several papers on nematodes where this appears to be the case, that despite treatment with tetracycline the infection cannot be completely cured. This is no surprise because for nematodes the wolbachia may be an obligate symbiont. This is not the case in Trichogramma, Tetracycline can completely cure the wasps of their infection.

> Similar to the obligate symbiont in nematodes, removing Wolbachia inhibited oogenesis in parasitic wasps (Dedeine et al., 2001; Wang et al., 2017). After removing Wolbachia with antibiotics, some Wolbachia-induced asexual strains cannot revert to normal sexual reproduction because of the loss of sexual functions in either male or female adults (Russell and Stouthamer, 2011; Wang et al., 2017).

> Wolbachia cannot be removed completely in T. pretisoum we tested, that is the different point we found comparing the previous reports. It had confused us for several years since we obtained the parasitoids from arbico-organic, USA in 2015. To remove Wolbachia, antibiotics including tetracycline, ciprofloxacin, sulfadiazine, and their mixtures were screened (Zhang et al., 2020), and an artificial host was developed to exclude contamination from host tissue (Zhu, 2019). Moreover, Wolbachia can be detected and localized on the parasitoids ’bodies by FISH after being fed with antibiotics (Guo, 2022).  

As more and more papers finding that the Wolbachia cannot be removed completely by antibiotics in wasps, weevils, and nematodes as listed in Table 1, I think the methods used to DNA extraction and detection make the different result. (1) The different method of extraction total DNA. Trichogramma is a tiny wasp less than 1 mm, traditional method used for DNA extractions generate partly DNA loss because they require several steps and may include transfers of DNA extracts to additional containers or washing/desalting procedures using various commercial filters or columns. We used the one-step to extract DNA, by which it do not require tube transfers. It is more efficient and suitable for indicidual Trichogramma DNA extraction because no DNA (Wolbachia and wasp) waste or loss. (2) Methods used to detect Wolbachia in last few years ago is no sensitive as Q-PCT or 16s rRNA-seq. (3) May be the resistant Wolbachia line.

Table 1 List of Wolbachia unremoved by tetracycline

speices

Host or no host

DNA extraction

Detection

Primers

reference

Trichogramma pretiosum

endoparasitoid

TIANamp Genomic DNA Kit

Q-PCR/

Q-PCR/

Nian et al, 2022

Trichogramma dendrolimi

endoparasitoid

Chelex-100

Q-PCR

wsp

Chen et al, 2022

Pantomorus postfasciatus

weevil

Salting -out protocol

Q-PCR

wsp

Rodriguero et a., 2020

Hypothenemus hampei

Coffee Berry Borer

DNeasy Plant Mini Kit

PCR

wsp

Mariño et al., 2017

Encarsia formosa

endoparasitoid

Wizard® SV Genomic DNA Purification System

PCR/Q-PCR/FISH

wsp/16S

Wang et al., 2017

  1. In addition, they show that the Trichogramma they rear on wolbachia free hosts still contain quite a community of bacteria, however the meaning of this is unclear because no control experiment was done by simply extracting the dna of the host eggs themselves. The bacteria found in the samples of the wasp may simply reflect the bacteria found in or on the eggs of the Corcyra.

>To reduce the background community of bacteria by the host, Corcyra cephalonica used in our experiment were fed with tetracycline hydrochloride for multiple generations (over two years). What we focus on is how Wolbachia manipulates bacterial diversity in its host. Before we use their eggs as hosts, we used PCR to confirm they are Wolbachia-free.

  1. While the authors collect a lot of data from these microbiomes, what are the conclusions? And why does it matter?

>Thanks for reminding us. We neglected to discuss the microbiomes in the previous manuscript. Here, a subtitle “Diversity changes of endosymbiotic bacterium” were discussed in reversion. We found the bacterial communities differed significantly between heredity and PI-Wolbachia thelytoky Trichogramma. The content of Wolbachia in the PI-Wolbachia thelytokous population is extremely high (up to 99%). Speculating that Wolbachia manipulates bacterial diversity in its host (Ourry et al., 2021), as it achieves the degree fixed in the host population (Russell et al., 2018). After treatment with antibiotics and high temperatures, the relative abundance of Wolbachia decreased to 20.12%, whereas it promoted the increase of co-bacterial diversity in Trichogramma. With the contents of Wolbachia dropping, the higher contents of male.

>Conclusion: Wolbachia density dependence drives the reproduction switch in T. pretiosum from thelytoky to arrhenotoky, once the density of Wolbachia is lower than the threshold thelytoky requires. Finding will extent the the knowledge about Wolbachia and endosymbiotic bacterium of host.

  1. The origin and identity of the Trichogramma used in the experiments are also unclear. When were these lines originally collected? The authors state their identity was confirmed using the sequence of ITS2 and blasting the sequences against genbank. There is no accession number given for the sequences they obtained, in GenBank there quite many sequences deposited with the wrong species name given to them.

> The information of Trichogarmma we used in the manuscript as showed in table 2. We also added the table in the new version. Both T. embryophagum and T. cacoeciae original from Germany and they were introduced from Dr. Sherif Hassan, Institute for Biological Pest Control, BBA, Germany, in 2003. T. pretiosum were buy in the arbico-organic, USA. We primitively keep the parasitoids in the Institute of Zoology, Guangzhong Academy of Sciences. For backup, wasps also rearing in my laboratory in South China Agricultural University. Species identified by male genitalia or its2 sequences (Table 2).

Table 2 Information of Trichogramma spp. tested in the study

Species

origin

Accession number

ID/ Reference

Genital capsule

T. pretiosum

arbico-organic, USA

MH890848

(Submitted in 2018)

99.57%/ Taxonomy ID: 7493 (Lindsey et al., 2018)

T. cacoeciae

Germany

MH890823

(Submitted in 2018)

100%/ EU547670, JF920445

98.26%/AY166700(Almeida and Stouthamer, 2003)

-

T. embryophagum

Germany

Thelytokous

MH890833(Submitted in 2018)

97.84% /JF920455 (Poorjavad et al., 2012)

-

Germany

Bisexual

AY244465(Submitted in 2003)

  1. In addition the literature is quite unclear of the difference between T. embryophagum and T. cacoeciae (see Sumer F, Tuncbilek AS, Oztemiz S, Rugman-Jones PF, & Stouthamer R (2009) A molecular key to the common species of Trichogramma of the Mediterranean region. Biocontrol 54(5):617-624.)

> Yes, T. embryophagum and T. cacoeciae are very closely related species. The first time we obtained these two species from Dr. Simon Grenier was in 1999. T. embryophagum is bisexual reproduction, and T. cacoeciae is thelytoky. So I finished the ITS2 sequencing of both species and the genital capsule of T. embryophagum in 2003, as shown in Table 2. The two species tested in the paper are thelytoky. Their ITS2 was sequenced and submitted in 2018. Afterwards, we check the ITS2 regularly to avoid species blending. However, its2 cannot distinguish closely related species because the genetic distances of its2 in interspecies are smaller than those of intraspecies. We used them as two species, following the name list when they were introduced to China.

Reviewer 2 Report

The differences in reproductive behavior and bacterial community diversity between two parthenogenesis modes of Trichogramma spp. under the influence of environmental factors are systematically studied in the paper. It was figured out that antibiotics cannot permanently deplete Wolbachia from their host, resulting in its rebounding. There are several points that needed to be improved before your manuscript has better shape to be published in the journal. There are several comments in the pdf file in the attachment. In summary, there are the following points need to respond:

1. The introduction is very general, should describe clearly the innovation of the article should be described clearly the innovation of the article. What scientific hypothesis is put forwarded.

2. The Wolbachia infection with negative "-" or 0% should be consistent among tables 1 to 3.

3. What are the differences of two types of thelytoky in the Trichogramma species? Tetracycline hydrochloride and high temperature can cause T. cacoeciae and T. embryophagum produce male offspring. As an antibiotic, why sulfadiazine could restore entirely arrhenotokous in Wolbachia infected T. pretiosum, while didn’t work for T. cacoeciae and T. embryophagum which caused thelytoky by heredity? Analyze in the discussion section.

4. Wolbachia can be detected in the male pretiosum. It is interesting! Most research focuses on Wolbachia from thelytokous female. Do you exclude the background, such as the host contaminant of Wolbachia?

5. What implications do the results have for further research?

Author Response

Dear Revivers,

We are thanks for the valuable suggestions in the referee’s report. We have made careful modifications on the manuscript. The point-by-point response was uploading with the manuscript.

Best Regards!

Yours sincerely,

Xiaofang

Response to comments 

  1. The introduction is very general, should describe clearly the innovation of the article should be described clearly the innovation of the article. What scientific hypothesis is put forwarded.

> In the revision, we put the results in more detail on the abstract. The study is the first to show the bacterial communities in parthenogenetic Trichogramma before and after antibiotic treatment.

  1. The Wolbachia infection with negative "-" or 0% should be consistent among tables 1 to 3.

>Thanks for reminding. I have modified it in the article.

  1. What are the differences of two types of thelytoky in the Trichogramma species? Tetracycline hydrochloride and high temperature can cause T. cacoeciae and T. embryophagum produce male offspring. As an antibiotic, why sulfadiazine could restore entirely arrhenotokous in Wolbachia infected T. pretiosum, while didn’t work for T. cacoeciae and T. embryophagum which caused thelytoky by heredity? Analyze in the discussion section.

> T. cacoeciae and T. embryophagum are thelytoky caused by heredity, and T. pretiosum is thelytoky caused by Wolbachia, which will be depleted by antibiotics. Once the titer of Wolbachia is less than the minimum threshold the thelytoky inquired, T. pretiosum will revert to the arrhenotoky.

> It is possible that the mechanisms of action of antibiotics (Kent, 2000; Castelli et al., 2001; Chopra et al., 2001) are different, or that the resistance of Trichogramma to different antibiotics (Nian et al., 2022). The reasons for the male production in the Wolbahica free Trichogramma when treated by tetracycline hydrochloride and high temperature may be because of genetic mechanisms (Jahner et al., 2015), or chemicals increasing the emergence rate of males (Leite et al., 2017a, b). The mechanism of sexually dimorphic traits in Trichogrmama is still scarce and should be studied further.

  1. Wolbachia can be detected in the male T. pretiosum. It is interesting! Most research focuses on Wolbachia from thelytokous female. Do you exclude the background, such as the host contaminant of Wolbachia?

> Yes, to reduce the background noise from the host, the hosts used in our experiment were fed with stetracycline hydrochloride successively for multiple generations (over two years). Wolbachia was freed by the 16S RNA sequencing of T. cacoeciae and T. embryophagum. It means the host and Trichogramma are free of Wolbachia.

  1. What implications do the results have for further research?

> Once we get the complete arrhenotokous T. pretiosum, we will explore how the Wolbachia regulate hosts between thelyotoky and arrhenotoky with the same genetic background. In the ongoing manuscript, the functions of the differentially expressed genes and the key gene involved in regulation were further verified.